# The Impact of 24-Month Etanercept Therapy on Changes in Adiponectin, Leptin and Tenascin C Levels in the Blood of Children with Juvenile Idiopathic Arthritis

**DOI:** 10.3390/ph18091423

**Published:** 2025-09-22

**Authors:** Jan Siwiec, Kornelia Kuźnik-Trocha, Katarzyna Winsz-Szczotka, Katarzyna Komosińska-Vassev, Andrzej Siwiec, Krystyna Olczyk

**Affiliations:** 1Clinical Department of Pediatrics, Faculty of Medical Sciences in Katowice, Medical University of Silesia in Katowice, Zapolskiej 3, 41-218 Sosnowiec, Poland; jan.siwiec@sum.edu.pl; 2Department of Clinical Chemistry and Laboratory Diagnostics, Faculty of Pharmaceutical Sciences in Sosnowiec, Medical University of Silesia in Katowice, Jedności 8, 41-200 Sosnowiec, Poland; winsz@sum.edu.pl (K.W.-S.); kvassev@sum.edu.pl (K.K.-V.); olczyk@sum.edu.pl (K.O.); 3Department of Developmental Age Physiotherapy, Faculty of Health Sciences in Katowice, Medical University of Silesia in Katowice, 40-752 Katowice, Poland; asiwiec@sum.edu.pl

**Keywords:** adiponectin, leptin, tenascin C, juvenile idiopathic arthritis, etanercept

## Abstract

**Background/Objectives**: The most commonly diagnosed group of rheumatic diseases in children is juvenile idiopathic arthritis. It is characterized by a chronic inflammatory process that leads to the degradation of the bone and joint system and increased secretion of pro-inflammatory cytokines such as TNF-α, IL-1, and IL-6. These cytokines contribute to the dysregulation of adipocytokine metabolism, including adiponectin and leptin, as well as extracellular matrix components, such as tenascin C. While it is known that children with JIA exhibit TNF-α-stimulated degradation of most ECM cartilage components, the effect of TNF-α antagonists, such as etanercept, on these processes has not yet been evaluated. Therefore, the aim of our study was to assess the dynamics of changes in tenascin C, adiponectin, and leptin levels in the blood of children with JIA, both before and during therapy. **Methods**: The study material consisted of blood samples collected from 66 children of both sexes, including 40 girls and 26 boys diagnosed with juvenile idiopathic arthritis and treated with etanercept, as well as from 40 healthy children (22 girls and 18 boys). The quantitative assessment of adiponectin, leptin, and tenascin C levels was performed using commercial ELISA tests. **Results**: The conducted study revealed that untreated children with JIA exhibit altered plasma levels of all examined parameters—adiponectin, leptin, and tenascin C. Specifically, there was an increase in adiponectin concentration and a decrease in leptin as well as TNC levels compared to healthy children. The results demonstrated the beneficial effects of the TNF-α antagonist, i.e., etanercept, which not only improved the clinical condition of children with JIA but also positively influenced the metabolism of both adipokines and tenascin C. **Conclusions**: The obtained results suggest the potential use of adiponectin, leptin, and tenascin C as biochemical markers of the effectiveness of etanercept therapy in inhibiting the progression of degenerative joint changes in children with JIA treated with TNF-α inhibitors.

## 1. Introduction

Juvenile idiopathic arthritis (JIA) is the most common autoimmune connective tissue disease among pediatric patients. Regardless of its form, JIA remains a condition whose progression can only be slowed. First-line therapy, which includes nonsteroidal anti-inflammatory drugs (NSAIDs) and disease-modifying antirheumatic drugs (DMARD), including methotrexate, sulfasalazine, or prednisone, does not always lead to clinical improvement. In such cases, patients over the age of four may qualify for biological therapy, including treatment with etanercept (ETA) [1,2,3,4]. However, despite ongoing treatment, the disease remains progressive, with periods of remission and exacerbation, ultimately leading to joint structure destruction. The progressive degradation of articular cartilage in JIA, initially manifesting as inflammation of the periarticular soft tissues, synovial hypertrophy, swelling, pain, and limited joint mobility, over time results in irreversible structural changes in the affected joints [2,5,6].

The pathogenesis of juvenile idiopathic arthritis (JIA) is driven by chronic inflammation resulting from immune system dysfunction [7]. Key immune cells—including T lymphocytes, macrophages, dendritic cells, and neutrophils—contribute to this process through excessive secretion of pro-inflammatory cytokines, especially TNF-α, IL-1, IL-6, and IL-18 [1,2,7,8]. TNF-α plays a central role by promoting leukocyte migration, stimulating T cell proliferation, inducing collagenase production, and activating osteoclasts, which together lead to cartilage damage and bone resorption. It also stimulates monocytes and macrophages to release additional pro-inflammatory mediators and reactive oxidative species [5,9,10]. In addition, adipose tissue contributes to immune regulation by secreting signaling molecules, protein factors, and hormones, including adiponectin (ADPN) and leptin (LEP) [11,12]. Initially, these adipocytokines were associated with the pathogenesis of eating disorders and related weight loss. However, given their ability to modulate immune and inflammatory responses, their involvement in the development of juvenile idiopathic arthritis appears undeniable [11,12,13,14]. These adipokines are engaged not only in the inflammatory process but also influence the expression of numerous extracellular matrix (ECM) components, including probably tenascin C (TNC).

Tenascin C is a high-molecular-weight glycoprotein, ranging from 200 to 400 kDa [15]. It plays a crucial role in the morphogenesis of limbs, kidneys, lungs, mammary glands, and teeth, as well as in cartilage and bone development [16,17]. In adults, TNC primarily functions in adhesion, contributing to tissue remodeling and repair by promoting the recruitment, migration, and differentiation of myofibroblasts to damaged sites TNC signaling is environment-dependent, and its activity relies on interactions with other extracellular matrix components [15,16,17,18]. While TNF-α-stimulated ECM degradation in JIA cartilage has been well documented, the impact of TNF-α antagonists, such as etanercept, on these processes, have not yet been fully assessed. Inflammation-induced metabolic changes in ECM components—including tenascin C—are influenced by cytokines and adipocytokines with opposing biological effects, such as adiponectin and leptin; therefore, their circulating levels may reflect these metabolic alterations. Although these molecules are known to participate in the pathogenesis of JIA, their clinical significance in for monitoring anticytokine therapy remains unclear. Therefore, the primary aim of this study was to evaluate the dynamics of changes in tenascin C, adiponectin, and leptin concentrations in the blood of children with JIA, both prior to and during etanercept therapy. Given the roles of adipokines in immune modulation and systemic inflammation and the function of tenascin C as an extracellular matrix glycoprotein involved in tissue remodeling and inflammatory responses, these markers may have potential value as biochemical indicators of therapeutic efficacy in JIA.

## 2. Results

### 2.1. Assessment of Adiponectin, Leptin and Tenascin C Concentrations in the Blood of Healthy Children and Children with Juvenile Idiopathic Arthritis

Data on plasma levels of adiponectin, leptin, and tenascin C in healthy children and those with JIA—both before the initiation of biological treatment and in the same children at the 24-month mark of etanercept therapy—are presented in Table 1.

The study revealed a significant (*p* < 0.005) increase in adiponectin concentration in the blood of children with juvenile idiopathic arthritis who had not undergone etanercept therapy compared to healthy children. Moreover, it was demonstrated that 24 months of ETA treatment significantly reduced plasma adiponectin concentration in JIA patients to values even lower than those observed in healthy children (*p* < 0.01). Statistical analysis of leptin w concentration showed a significant (*p* < 0.05) decrease in this adipocytokine in untreated JIA patients compared to healthy children. However, etanercept therapy led to an increase in leptin levels in JIA patients, bringing them closer to those observed in healthy children. Furthermore, as shown in Table 1, plasma tenascin C concentration in JIA patients were significantly lower than in healthy children, both before and after 24 months of etanercept therapy. Although biological treatment, which contributed to clinical improvement in patients, also led to an increase in TNC concentration, it did not fully normalize this parameter within the two-year treatment period.

### 2.2. The Effect of Gender on Plasma Concentrations of Adiponectin, Leptin, and Tenascin C in Healthy Children and Children with Juvenile Idiopathic Arthritis

The analysis of the influence of sex on adiponectin levels showed no significant differences in plasma concentrations of this adipocytokine between girls and boys (*p* > 0.05) across all study groups, including healthy children and JIA patients, both before and after 24 months of etanercept therapy. However, significant sex-related differences were observed in plasma levels of both leptin and tenascin C, but only between healthy girls and boys (*p* < 0.05) (Table 1).

### 2.3. The Effect of Etanercept Therapy on Plasma Concentrations of Adiponectin, Leptin, and Tenascin C in Children with Juvenile Idiopathic Arthritis

As a part of this study, the dynamics of changes in the concentrations of the examined markers in the blood of children with JIA during biological therapy were assessed. The initial value was defined as the plasma concentration of the examined adipokines in sick children before the first dose of the TNF-α inhibitor (T0), while subsequent values corresponded to plasma concentrations obtained at the different time points during etanercept therapy, specifically at the third (T3), sixth (T6), twelfth (T12), eighteenth (T18), and twenty-fourth (T24) month of treatment. The plasma concentrations of adiponectin, leptin, and tenascin C in children with JIA during therapy are presented in Figure 1.

As shown in Figure 1, the applied biological therapy significantly influences the metabolism of the evaluated cytokines derived from adipose tissue, which is reflected in changes in adiponectin concentration in the blood of children with JIA at different stages of treatment. A gradual, progressive decrease in adiponectin concentration in the blood of children with JIA was observed, continuing up to the twenty-fourth month of therapy. This progressive reduction in plasma concentration of the assessed adipocytokine was statistically significant, starting from the twelfth month of treatment (*p* < 0.0033).

The analysis of the effect of etanercept therapy on changes in plasma leptin levels in children with juvenile idiopathic arthritis over successive months of treatment revealed a gradual increase in the concentration of this adipocytokine in the blood of sick children observed up to the twenty-fourth month of therapy. The differences in plasma leptin concentrations during etanercept treatment were statistically significant from the twelfth month of administration, as illustrated in Figure 1b.

In contrast, the evaluation of the effect of etanercept therapy on tenascin C blood concentration in children with JIA over successive periods of administration showed that TNC concentration significantly increased only after 24 months of therapy (*p* < 0.05) compared to tenascin C levels observed in both untreated children and those undergoing 3-, 6-, 12-, and 18-month therapy, as illustrated in Figure 1c.

In our study, all children treated with etanercept demonstrated marked clinical improvement, as reflected by very low JADAS scores after 24 months of therapy. This clinical improvement was accompanied by significant changes in the levels of the studied biomarkers, i.e., ADPN, LEP, and TNC. These results indicate that the therapeutic effect of etanercept, confirmed by improved JADAS scores, is accompanied by changes in adipokine and tenascin C profiles toward levels observed in healthy individuals, highlighting the potential of these biomarkers for monitoring the response to JIA treatment.

### 2.4. Multiple Linear Regression Model

We performed a multivariate multiple regression analysis to assess the influence of age, sex, BMI, and disease duration on changes in plasma concentrations of tenascin C, adiponectin, and leptin in patients with JIA undergoing etanercept (ETA) treatment. For tenascin C, the overall model was not statistically significant (F(4.61) = 1.30; *p* = 0.28), and the predictors accounted for only 7.8% of the variance in TNC change (R^2^ = 0.078; adjusted R^2^ = 0.018). These results indicate that none of the studied variables had a statistically significant effect on the change in tenascin C levels after treatment. Similar findings were obtained for both adipocytokines studied—adiponectin (F(4.61) = 0.36; *p* = 0.84; R^2^ = 0.023) and leptin (F(4.61) = 0.40; *p* = 0.81; R^2^ = 0.026).

### 2.5. Correlation Analyses

As a part of the study, an assessment was also conducted to evaluate the correlation between the examined biological markers—tenascin C and adiponectin and leptin concentrations—in the blood of children with juvenile idiopathic arthritis before the initiation of etanercept therapy, as well as at various time points during treatment, specifically at 3, 6, 12, 18, and 24 months of therapy. The analysis revealed weak positive correlations between tenascin C and leptin concentration (R = 0.261; *p* < 0.05), as well as between tenascin C and adiponectin concentration (R = 0.252; *p* < 0.05) in the blood of untreated children with JIA. These relationships are illustrated in the scatter plots shown in Figure 2 and Figure 3, respectively.

The analysis conducted in this study regarding the relationship between tenascin C and adiponectin concentrations in the blood of children with juvenile idiopathic arthritis during successive months of etanercept therapy—specifically in the third (R = 0.016; *p* > 0.05), sixth (R = −0.065; *p* > 0.05), twelfth (R = −0.061; *p* > 0.05), and eighteenth (R = −0.216; *p* > 0.05) months—did not reveal any significant associations between these parameters. However, a moderate negative correlation (R = −0.244; *p* < 0.05) was observed between the extracellular matrix protein and adiponectin concentrations in the blood of JIA patients at the 24th month of biological therapy, as illustrated in Figure 4.

In contrast, the correlation analysis between tenascin C and leptin concentrations in the blood of children with juvenile idiopathic arthritis undergoing biological therapy for two years—until clinical improvement was achieved—did not reveal any significant associations between these parameters at any assessed time points during treatment. This includes the third (R = 0.120; *p* > 0.05), sixth (R = 0.231; *p* > 0.05), twelfth (R = 0.205; *p* > 0.05), eighteenth (R = 0.116; *p* > 0.05), and twenty-fourth month (R = 0.153; *p* > 0.05) of therapy.

For a more comprehensive assessment of the biochemical changes occurring in the course of juvenile idiopathic arthritis, an analysis was also conducted to evaluate the relationships between adiponectin, leptin, and tenascin C concentrations and nonspecific inflammatory markers, namely erythrocyte sedimentation rate (ESR) and C-reactive protein (CRP) in the blood of children with JIA, both untreated and undergoing biological therapy. The obtained results are presented in Table 2.

As shown in the data presented in the table, no relationship was found between the concentrations of the evaluated markers, i.e., ADPN, LEP, and TNC, and the CRP level or ESR value in the blood of children with juvenile idiopathic arthritis, either before the administration of the biological drug or after two years of its use.

## 3. Discussion

Adipose tissue contributes to immune regulation by secreting proteins involved in immunoinflammatory responses, including adiponectin and leptin [12]. Although traditionally considered an anti-inflammatory molecule, adiponectin can exert pro-inflammatory effects under certain conditions [19]. In vitro studies have shown that it stimulates chondrocytes and synovial fibroblasts to produce matrix-degrading enzymes (e.g., MMP-3, MMP-10) and pro-inflammatory cytokines, including TNF-α [20]. Li et al. [20] demonstrated that all adiponectin isoforms can induce the synthesis of matrix metalloproteinases, with effects proportional to adiponectin concentration. Elevated ADPN levels have been reported in children with newly diagnosed JIA [13], and our findings also indicate increased adiponectin concentrations in children eligible for etanercept treatment due to poor response to conventional therapy. Furthermore, we observed correlations between adiponectinemia and markers of cartilage degradation, such as COMP and YKL-40. However, Ilisson et al. [21] found no difference in ADPN levels between untreated JIA patients and healthy controls. These discrepancies highlight the need for further studies focusing on individual adiponectin isoforms. The mechanisms underlying increased adiponectin levels in JIA are likely multifactorial and may contribute to disease progression. Its pro-inflammatory activity may be related to COX-2 stimulation, which promotes prostanoid synthesis and sustains inflammation [22,23]. This mechanism is supported by studies in patients with severe RA—a disease sharing clinical features with JIA— that examined the relationship between inflammation and adiponectin levels [19,24]. Gonzalez-Gay et al. [24] found that in RA patients, severe inflammation was independently and negatively correlated with circulating adiponectin, whereas Minamino et al. [19] reported a positive correlation between adiponectin levels and disease activity. However, neither study confirmed a significant effect of TNF-α inhibitor therapy on circulating adiponectin levels in RA [19,24]. Although our study did not demonstrate a correlation between plasma ADPN levels and non-specific inflammatory markers (CRP, ESR), the significant reduction in adiponectinemia following etanercept therapy supports the role of inflammation in regulating this adipokine’s metabolism. Our findings suggest that ADPN levels may serve as a potential biomarker for monitoring the efficacy of biological therapy in JIA.

In the case of the second examined adipokine, leptin, a significant reduction in its concentration was observed in the blood of children with JIA before the initiation of etanercept treatment, compared to healthy children. These results are consistent with our previous studies, which also demonstrated that leptin levels in the blood of children, particularly girls, with newly diagnosed JIA were significantly lower than those in the control group [13]. It has been suggested that at the onset of clinical symptoms, the pool of adipocytes synthesizing adipokines is significantly reduced (with BMI comparable to or even lower than in healthy children), and that leptin synthesis by non-adipose tissue cells is insufficient to normalize leptinemia in children with newly diagnosed, untreated JIA [25]. Our studies showed that etanercept therapy, which improved the clinical condition of patients (as reflected by low JADAS scores), also influenced blood leptin concentrations, leading to their normalization. Gonzalez-Gay et al. [26] reported no effect of anti-TNF therapy on leptinemia in patients with severe RA during TNF-α blockade. Moreover, these authors demonstrated that circulating leptin levels were not associated with disease activity but rather reflected obesity. Although our study did not reveal any effect of BMI on blood leptin levels in children with JIA, either before or after 24 months of therapy, we cannot exclude the possibility that an increasing adipocyte pool contributed to changes in leptin levels during etanercept treatment. However, neither the study by Gonzalez-Gay et al. [26] nor ours demonstrated an association between leptin levels and inflammatory markers such as CRP and ESR.

The therapy used in JIA patients with TNF-α receptor blockers in the form of a biological drug, while inhibiting the inflammatory process involving this factor and leading to clinical improvement, does not completely eliminate joint inflammation. This ongoing process is sustained by various cytokines, particularly IL-1, IL-6, and other inflammatory mediators and growth factors [1,4,7]. These pro-inflammatory cytokines influence the expression of many extracellular matrix components, including tenascin C, which also exhibits pro- inflammatory properties. According to the study by Shukla A et al. [27], children with active JIA have elevated serum TNC levels compared to healthy children. These authors also demonstrated that regular NSAID therapy reduces TNC levels. The results of our study, which demonstrated very low concentrations of tenascin C in the blood of children with JIA before biological therapy, but previously treated with disease-modifying antirheumatic drugs, partially correspond with the results of Shukla et al. [27]. However, during etanercept therapy, a progressive increase in TNC levels was observed, reaching a value by the 24th month of treatment—when patients experienced significant clinical improvement, as reflected by very low JADAS scores—that was still lower than the level found in healthy children. This could likely result from the suppression of inflammation due to etanercept therapy, while the repair process of damaged joint structures is strongly induced. It is well known that excessive tenascin C gene expression always occurs during wound healing and tissue remodeling, and this glycoprotein is often used as a biochemical marker of successful tissue repair [28,29]. The effect of ETA therapy on TNC levels in children with JIA has not been previously evaluated, making it impossible to compare the results of this study with those of other researchers. However, there are reports regarding the influence of biological drugs on TNC levels in rheumatoid arthritis. Page et al. [30] found that treating patients with early RA using infliximab combined with methotrexate led to a transient decrease in blood TNC levels during the first year of therapy, followed by an increase after approximately one year of continued treatment. These researchers did not find a correlation between serum TNC levels and clinical markers of inflammation related to RA activity. However, their study confirmed the usefulness of TNC level assessment as a good indicator of erosive joint damage [30]. These findings align with the analyses conducted in this study regarding the relationship between plasma TNC levels and non-specific inflammatory markers such as CRP and ESR. No correlation between the evaluated parameters was confirmed in children with JIA, whether untreated or undergoing ETA therapy.

However, some limitations of the present study must be acknowledged, particularly the relatively small sample size, which restricts the generalizability of our findings. Another limitation is the absence of pubertal status assessment, which may have influenced adipokine levels through hormonal changes associated with puberty. Given the participants’ age range (4 to 16 years), it is likely that some were undergoing pubertal transition, potentially introducing variability into the results. Nevertheless, most children were prepubertal at the time of etanercept initiation. Additionally, the lack of a control group receiving MTX monotherapy and the absence of a comparison group treated with other biologic agents represents another limitation of this study. Importantly, the absence of comparator biologic agents limits the generalizability of our findings to etanercept and precludes direct comparisons with other biologic treatments.

## 4. Materials and Methods

### 4.1. Subjects

The study included 66 children of both sexes—40 girls and 26 boys—aged 4 to 16 years, diagnosed with juvenile idiopathic arthritis, according to the criteria of the International League of Associations for Rheumatology (ILAR) [31], based on a minimum disease duration of six weeks and clinical symptoms such as joint pain and swelling, limited range of motion, and growth disorders. Diagnosis was confirmed based on laboratory parameters obtained from blood tests, including the assessment of non-specific inflammatory markers such as CRP concentration and ESR values, as well as the presence of RF factor and ANA antibodies. The disease activity in the examined children was assessed using the JADAS27 (Juvenile Arthritis Disease Activity Score 27), which ranges from 0 to 57 and includes four variables: the physician’s global assessment of disease activity, the child’s/parent’s assessment of well-being, the number of active joints (evaluating 27 joints), and the ESR value [32]. Based on the obtained results, children with JIA were classified for appropriate therapy, which included the use of sulfasalazine (at a dose of 25 mg/kg body weight per day), prednisone (at a maximum dose of 1 mg/kg body weight per day with gradual dose reduction), and methotrexate (10–20 mg/m^2^ body surface area per week). Children with JIA who did not show clinical improvement despite conventional treatment with a combination of two disease-modifying drugs/immunosuppressants at recommended doses (including methotrexate) for three to six months were qualified for biological therapy with etanercept under the Polish Therapeutic Program [33]. Patients over four years old who met the criteria for polyarticular JIA—with at least five swollen joints and at least three joints with limited mobility and pain, as well as elevated ESR or CRP levels and a disease activity score of at least 4 out of 10 as assessed by the physician—were enrolled in the TNF inhibitor program (etanercept), designated as B.33. Additionally, children with extended or persistent oligoarticular JIA for over six months, presenting poor prognostic factors (according to American College of Radiology—ACR) and at least two swollen or restricted joints with pain, as well as a disease activity score of at least 5 out of 10, were also included [34]. The exclusion criteria were other forms of JIA and other chronic and autoimmune diseases, previous treatment with biologic drugs, and discontinuation of biologic therapy during the study period. Biological treatment involved administering etanercept via subcutaneous injections twice a week at 3–4-day intervals, at a dose of 0.4 mg/kg body weight (up to a maximum dose of 25 mg) or 0.8 mg/kg body weight (up to a maximum dose of 50 mg) once a week. In all patients, ETA was used in combination with methotrexate, sulfasalazine, and prednisone, with the latter two being discontinued after three months of effective therapy. Treatment was conducted for 24 months. The two-year biological therapy in all studied children resulted in clinical improvement, assessed based on ACR Pediatric 30 criteria. Improvement in children treated with ETA was confirmed when, after at least 11 months of biological drug administration (on average 11.26 ± 0.75 months after treatment initiation), there was no active synovial inflammation or extra-articular clinical symptoms of the disease, CRP and ESR levels were within reference ranges, the physician’s global assessment of disease activity improved, and morning stiffness lasted less than 15 min.

The control group consisted of 40 healthy children (22 girls and 18 boys) within an appropriately matched age range, who underwent scheduled preventive examinations. Children were included in the reference group if their routine diagnostic test results—such as blood morphology and ESR, as well as cholesterol, glucose, creatinine, and C-reactive protein levels—fell within the reference ranges for their age group. Additionally, children in the reference group had not experienced any illnesses requiring hospitalization, had not undergone any surgical procedures, and had not been on long-term pharmacological treatment in the year preceding the study.

In all children who participated in the study, both healthy and those with JIA, body weight and height were assessed, and the obtained results were used to calculate the BMI index. The demographic, anthropometric, and clinical characteristics of both healthy children and those with JIA—both before the administration of the biological drug and after the 2-year etanercept therapy—are presented in Table 3. Children with overweight, obesity, and other features of metabolic syndrome were excluded from the study.

The parents/legal guardians of both healthy children and those with JIA provided consent for the collection and use of biological material (venous blood) remaining after the required diagnostic tests. The study was conducted in accordance with the Declaration of Helsinki, and the protocol was approved by the Local Bioethics Committee of the Silesian Medical University in Katowice (KNW/0022/KB/168/18).

### 4.2. Biochemical Studies

Blood drawn from the cubital vein into tubes containing lithium heparin was centrifuged for 10 min at 1500× *g* at 4 °C. The plasma obtained in this manner was used for the required diagnostic tests, while the remaining portion was frozen and stored at −80 °C until the start of the study.

Quantitative assessments of adiponectin, leptin, and tenascin C concentration in the blood of children from both the control and study groups were performed on the same day, ensuring that inter-assay variability was negligible. In children with JIA, the concentrations of adiponectin, leptin, and tenascin C were measured before the administration of the first dose of the biological drug (T0) and at subsequent time points during etanercept treatment, following the schedule: third (T3), sixth (T6), twelfth (T12), eighteenth (T18), and twenty-fourth (T24) months of therapy.

Enzymatic immunoassays (ELISA) were used to quantify ADPN, LEP, and TNC, following the manufacturer’s protocol. ELISA kits dedicated exclusively to scientific research were used. Plasma ADPN concentration was assessed using the high-sensitivity Human Adiponectin ELISA Kit, High Sensitivity from BioVendor (Brno, Czech Republic), with a minimum detection limit of 0.47 ng/mL. Plasma LEP concentration was measured with the Human Leptin ELISA Kit from BioVendor (Brno, Czech Republic), with a minimum detection limit of 0.2 ng/mL. For the quantitative assessment of TNC concentration, the ELISA Kit for Tenascin C from Cloud-Clone Corp. (Houston, TX, USA) was used, with a minimum detection limit of 1.25 ng/mL. For all tested parameters, intra-assay variability was less than 8%.

### 4.3. Statistical Analysis

The obtained results were subjected to statistical analysis using the Statistica 13.3 software package (TIBCO Software Inc., Kraków, Poland). The analysis included the following steps: assessment of normality for each variable using the Shapiro–Wilk test, assessment of variance equality using the Snedecor-Fisher test, descriptive statistics for normally distributed variables presented as mean ± standard deviation, descriptive statistics for non-normally distributed variables presented as median (Me), interquartile range—lower quartile (Q_1_), and upper quartile (Q_3_), significance testing of differences between the control group and study groups using the Mann–Whitney U test, significance testing of differences between the study group at baseline (T0) and after 24 months of treatment (T24) using the Wilcoxon signed-rank test, significance testing of differences between multiple study group time points (various months of treatment) using the Friedman test for dependent samples and post hoc multiple comparisons. Assessment of correlation strength between two variables using Spearman’s rank correlation coefficient (R). A *p*-value < 0.05 was considered statistically significant for all applied tests and statistical analyses. In the case of multiple comparisons following the Friedman test, a Bonferroni correction was applied for the post hoc analyses, with the adjusted significance level set at *p* < 0.0033.

## 5. Conclusions

In children with active juvenile idiopathic arthritis, alterations in adiponectin, leptin, and tenascin C were observed, characterized by increased plasma ADPN and decreased LEP and TNC plasma concentrations, suggesting their involvement in the pathogenesis of JIA. Etanercept therapy, which improved the clinical status of patients, also influenced these molecules, leading to a significant reduction in adiponectin, an increase in tenascin C, and normalization of leptin levels. These findings indicate that changes in the levels of adiponectin, leptin, and TNC may reflect treatment response; however, further validation through correlations with standardized clinical and imaging-based measures is required before their clinical utility as biomarkers can be confirmed. The identification of reliable blood biomarkers to monitor therapeutic efficacy could ultimately provide a valuable complement or alternative to current clinical testing methods.

## Figures and Tables

**Figure 1 pharmaceuticals-18-01423-f001:**
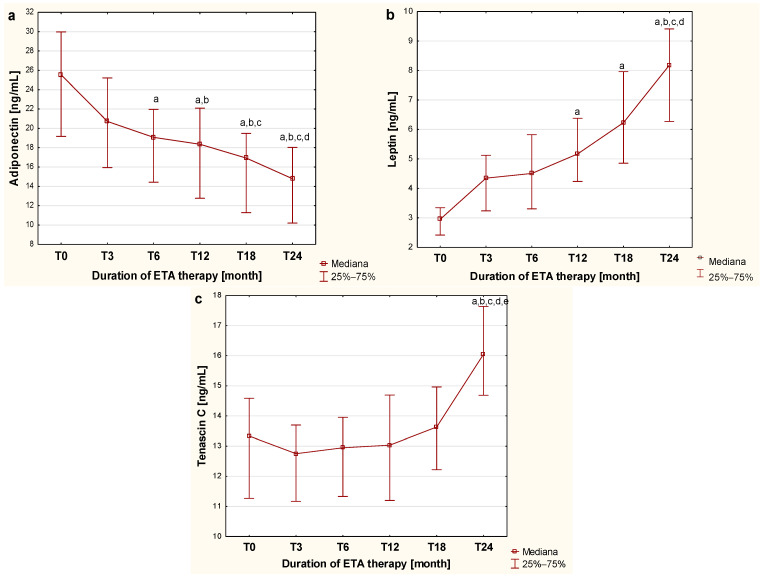
Dynamics of changes in adiponectin (**a**), leptin (**b**) and tenascin C (**c**) concentration in the blood of children with JIA before and during two-year etanercept therapy. a—statistically significant difference compared to group T0 (*p* < 0.0033); b—statistically significant difference compared to the T3 group (*p* < 0.0033); c—statistically significant difference compared to the T6 group (*p* < 0.0033); d—statistically significant difference compared to group T12 (*p* < 0.0033); e—statistically significant difference compared to group T18 (*p* < 0.0033).

**Figure 2 pharmaceuticals-18-01423-f002:**
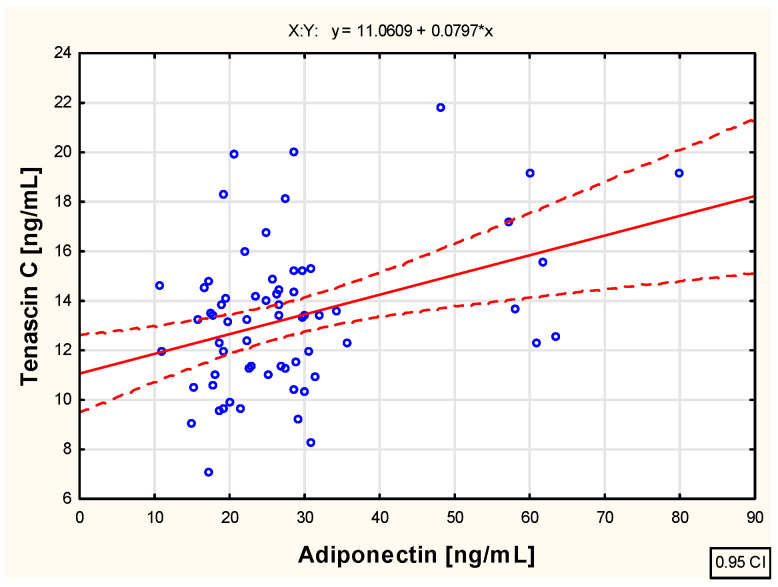
The relationship between the tenascin C and adiponectin concentration in the blood of children with juvenile idiopathic arthritis who are not biologically treated.

**Figure 3 pharmaceuticals-18-01423-f003:**
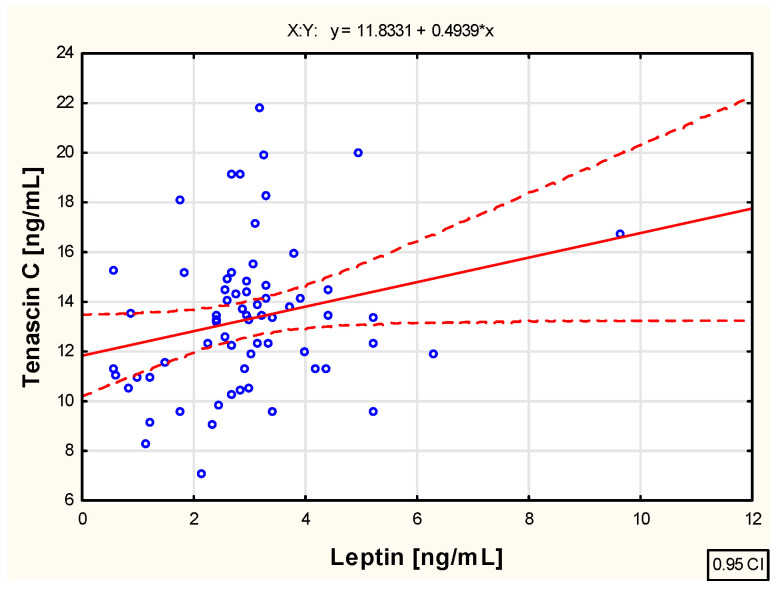
The relationship between the tenascin C and leptin concentration in the blood of not biologically treated children with juvenile idiopathic arthritis.

**Figure 4 pharmaceuticals-18-01423-f004:**
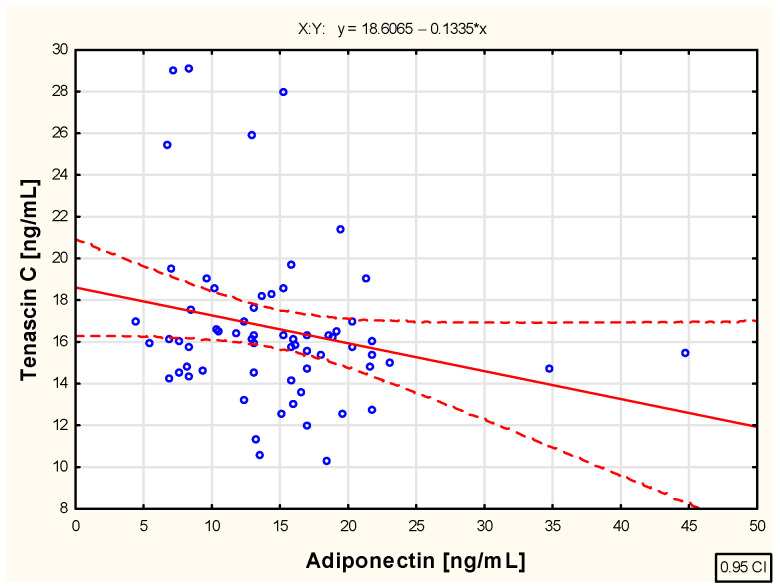
The relationship between tenascin C and adiponectin concentration in the blood of children with juvenile idiopathic arthritis treated with etanercept for 24 months.

**Table 1 pharmaceuticals-18-01423-t001:** Plasma concentrations of adiponectin, leptin, and tenascin C in healthy children and those with JIA, both before treatment and after 24 months of etanercept therapy.

Study Group		Adiponectin [ng/mL]	Leptin [ng/mL]	Tenascin C [ng/mL]
**Healthy children** (HC)	All children (n = 40)	19.64 (Q_1_ −16.11; Q_3_ −22.87)	5.78 (Q_1_ −3.15; Q_3_ −11.47)	17.87 (Q_1_ −16.53; Q_3_ −19.44)
	Girls (n = 22)	19.13 (Q_1_ −13.58; Q_3_ −21.77)	4.22 (Q_1_ −3.07; Q_3_ −7.02)	17.45 (Q_1_ −16.06; Q_3_ −18.04)
	Boys (n = 18)	20.01 (Q_1_ −16.67; Q_3_ −24.56)	10.28 ^b^ (Q_1_ −4.76; Q_3_ −14.77)	18.44 ^b^ (Q_1_ −17.45; Q_3_ −19.50)
**Children with JIA** (T0)	All children (n = 66)	25.54 ^a^ (Q_1_ −19.18; Q_3_ −29.98)	2.95 ^a^ (Q_1_ −2.42; Q_3_ −3.34)	13.33 ^a^ (Q_1_ −11.27; Q_3_ −14.59)
	Girls (n = 40)	26.82 (Q_1_ −19.20; Q_3_ −30.83)	2.96 (Q_1_ −2.51; Q_3_ −3.78)	13.37 (Q_1_ −11.28; Q_3_ −15.34)
	Boys (n = 26)	23.02 (Q_1_ −18.79; Q_3_ −28.99)	2.92 (Q_1_ −1.51; Q_3_ −3.31)	13.28 (Q_1_ −11.00; Q_3_ −14.09)
**Children with JIA after 24 months of** **therapy** (T24)	All children (n = 66)	14.80 ^a,c^ (Q_1_ −10.21; Q_3_ −18.03)	8.18 ^c^ (Q_1_ −6.27; Q_3_ −9.41)	16.05 ^a,c^ (Q_1_ −14.69; Q_3_ −17.64)
	Girls (n = 40)	15.84 (Q_1_ −13.08; Q_3_ −18.77)	8.26 (Q_1_ −6.85; Q_3_ −9.42)	15.95 (Q_1_ −14.72; Q_3_ −16.66)
	Boys (n = 26)	12.69 (Q_1_ −8.46; Q_3_ −15.97)	8.05 (Q_1_ −5.22; Q_3_ −8.70)	16.45 (Q_1_ −14.27; Q_3_ −18.27)

The results are presented as median and interquartile range (Q_1_—lower quartile and Q_3_—upper quartile); ^a^—statistically significant difference compared to the control group (*p* < 0.01); ^b^—statistically significant difference compared to healthy girls (*p* < 0.05); ^c^—statistically significant difference compared to children before treatment (*p* < 0.000005).

**Table 2 pharmaceuticals-18-01423-t002:** Relationship between plasma concentrations of adiponectin, leptin, and tenascin C and inflammatory parameters.

Parameter Tested		CRP (mg/L)		ESR (mm/h)	
		T0	T24	T0	T24
**Adiponectin** [ng/mL]	R	0.055	−0.027	0.067	−0.074
	*p*	0.660	0.827	0.591	0.555
**Leptin** [ng/mL]	R	−0.142	0.026	−0.023	0.146
	*p*	0.255	0.834	0.853	0.242
**Tenascin C** [ng/mL]	R	−0.201	0.081	0.003	0.063
	*p*	0.105	0.520	0.981	0.620

Children with JIA before treatment (T0); Children with JIA after 24 months of therapy (T24).

**Table 3 pharmaceuticals-18-01423-t003:** Demographic and clinical characteristics of healthy children and children with JIA, before biological treatment and after 24 months of etanercept therapy (results presented as mean and standard deviation).

Parameter	Healthy Children	Children with JIA	
	HC (n = 40)	T0 (n = 66)	T24 (n = 66)
Age (years)	10.00 (7.50–12.50) *	11.00 (6.00–13.00) *	13.00 (8.00–15.00) *
Gender (girls/boys)	22/18	40/26	40/26
BMI (kg/m^2^)	20.05 ± 4.40	18.69 ± 4.32	19.42 ± 4.31
WBC (10^3^/μL)	6.91 ± 2.26	8.17 ± 2.60	8.04 ± 3.60
RBC (10^6^/μL)	4.66 ± 0.37	4.41 ± 0.42	4.59 ± 0.45
Hb (g/dL)	13.26 ± 1.03	12.33 ± 1.23	12.88 ± 1.48
Ht (%)	39.25 ± 2.93	37.07 ± 3.57	38.16 ± 3.70
PLT (10^3^/μL)	325.13 ± 66.39	333.44 ± 118.38	338.32 ± 92.63
TCH (mg/dL)	139.46 ± 16.91	159.30 ± 41.63	163.41 ± 25.73
Glucose (mg/dL)	82.53 ± 7.48	88.66 ± 16.67	92.92 ± 9.00
Cr (mg/dL)	0.43 ± 0.14	0.47 ± 0.15	0.50 ± 0.17
CRP (mg/L)	1.83 (1.03–2.14) *	5.17 (1.29–15.56) *	1.74 (0.46–6.08) *
ESR (mm/h)	6.00 (4.50–8.00) *	21.00 (10.00–34.00) *	14.50 (7.00–24.00) *
ANA	-	47% (positive)	47% (positive)
RF	-	100% (negative)	100% (negative)
JIA subtype	-	oligoarticular (34)/polyarticular (32)	oligoarticular (34)/polyarticular (32)
Duration of the disease (month)	-	4.59 ± 1.51	28.59 ± 1.51
JADAS-27	-	40.00 (36.00–48.50) *	1.00 (0.00–1.00) *
Drugs	-	MTX, EC, SSD	ETA, MTX

Results are expressed as mean ± SD; * results are presented in the form of median and quartile range (Q_1_—lower quartile and Q_3_—upper quartile); BMI, body mass index HC, healthy controls; WBC, white blood cell; RBC, red blood cell; Hb, hemoglobin; Ht, hematocrit; PLT, platelet; TCH, total cholesterol; Cr, creatinine; CRP, C-reactive protein; ESR, erythrocyte sedimentation rate; ANA, antinuclear antibodies; RF, rheumatoid factor.

## Data Availability

The original contributions presented in this study are included in the article. Further inquiries can be directed to the corresponding author.

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
