# Peer review of "The Impact of 24-Month Etanercept Therapy on Changes in Adiponectin, Leptin and Tenascin C Levels in the Blood of Children with Juvenile Idiopathic Arthritis"

_pharmaceuticals, 2025, doi:10.3390/ph18091423_

Round 1
Reviewer 1 Report
Comments and Suggestions for Authors
This study investigates longitudinal changes in circulating adiponectin, leptin, and tenascin C in a cohort of children with JIA treated with etanercept. The authors report increased adiponectin and decreased leptin and TNC at baseline compared with controls and describe normalization or partial recovery of these parameters after 24 months of etanercept therapy. The study addresses an important and relatively unexplored topic, and the longitudinal design is a strength. However, several aspects of the manuscript require clarification, further justification, or tempering of the conclusions. Please, see my comments below.
Major Comments
The heterogeneity of the JIA cohort (different ILAR subtypes, disease duration before etanercept, prior DMARD exposure) is not described in detail. Such information is critical, as disease subtype and treatment history may strongly influence biomarker levels.
The rationale for selecting adiponectin, leptin, and TNC as candidate biomarkers of treatment response is insufficiently justified. The introduction should better frame their translational relevance for clinical monitoring.
The claim that these molecules may serve as “biochemical markers of the effectiveness of etanercept therapy” is premature. The study does not provide correlations with validated clinical outcomes such as JADAS27, ACR Pedi 30/50/70 responses, or imaging-based markers. Without such associations, the clinical utility of these markers remains speculative.
The observed changes may reflect confounding factors (e.g., BMI, growth/puberty, combination therapy with MTX) rather than a direct effect of TNF blockade. This should be acknowledged more explicitly.
Multiple longitudinal comparisons were performed without adjustment for multiple testing. This raises the risk of type I error inflation. The authors should discuss this limitation and, ideally, provide corrected p-values.
No multivariable analyses were conducted to account for age, sex, BMI, or disease duration. Such analyses would strengthen the robustness of the conclusions.
The discussion briefly mentions the small sample size, but other important limitations should be emphasized: lack of comparator group treated with other biologics, potential confounding by concomitant methotrexate, and absence of repeated disease activity measurements.
The influence of puberty on adipokine levels is a particularly important confounder in a pediatric cohort spanning ages 3–19 years.
Minor Comments
The introduction and discussion are overly long and contain repetitions. They could be streamlined to improve readability.
Some sentences are cumbersome (e.g., lines 246–255). Shorter, more direct phrasing is recommended.
Figure 1: The statistical annotations (a, b, c, d, e) are not self-explanatory. Please expand the figure legend for clarity.
Tables: numerical formatting should be standardized (use either dot or comma consistently for decimals).
Several references are outdated; more recent literature on biomarker research in JIA should be incorporated (e.g., proteomic or multi-omic approaches to treatment response).
Ensure uniform formatting of DOIs (currently inconsistent in some entries).
The statement “Which could help monitor…” (lines 433–434) contains a grammatical error and should be corrected.
Conclusions should be tempered to emphasize hypothesis-generating findings rather than definitive biomarker validation.
Overall recommendation
The manuscript provides novel longitudinal data on circulating adipokines and tenascin C in JIA patients treated with etanercept. The results are of interest but currently overinterpreted.
The authors should temper their conclusions, provide clearer justification of biomarker selection, strengthen the discussion of limitations, and improve statistical reporting before the manuscript can be considered for publication.
Comments on the Quality of English LanguageSome sentences are cumbersome (e.g., lines 246–255). Shorter, more direct phrasing is recommended.
Author Response
Answer to the Reviewer's comment 1
We would like to thank the Reviewer for the evaluation of our article. We are grateful for the important comments, which we addressed in the revised manuscript. The text of our manuscript (in each individual part, changes made in red) has been modified, so as to facilitate its understanding and make it acceptable for publication.
Major Comments
- The heterogeneity of the JIA cohort (different ILAR subtypes, disease duration before etanercept, prior DMARD exposure) is not described in detail. Such information is critical, as disease subtype and treatment history may strongly influence biomarker levels.
ANSWER:
We thank the Reviewer for this important observation. The following information has been added to the text:
- ILAR subtypes:
- Exclusion criteria were clarified:
„Exclusion criteria included other forms of JIA and other chronic and autoimmune diseases, prior treatment with biologics, and discontinuation of biologic therapy during the study period” (lines 362–364).
- Detailed information on JIA subtypes has also been added to Table 3.
- Disease duration before etanercept and prior DMARD exposure:
- We have expanded the methods to include the following statement: „Children with JIA who did not show clinical improvement despite conventional treatment with a combination of two disease-modifying drugs/immunosuppressants at recommended doses (including methotrexate) for three to six months were qualified for biological therapy with etanercept under the Polish Therapeutic Program” (lines 351–354).
- Information on disease duration before initiation of etanercept and previous exposure to DMARD has been added to Table 3.
- The rationale for selecting adiponectin, leptin, and TNC as candidate biomarkers of treatment response is insufficiently justified. The introduction should better frame their translational relevance for clinical monitoring.
ANSWER:
Thank you for this valuable comment. We agree that a clearer justification for the selection of adiponectin, leptin, and tenascin C (TNC) as candidate biomarkers is essential.
In the revised manuscript, we have modified the Introduction to better highlight the translational and clinical relevance of these molecules (lines 92-105). Specifically, we now emphasize that adiponectin and leptin are key adipokines involved in immune modulation and systemic inflammation, both of which are relevant to the pathophysiology of juvenile idiopathic arthritis (JIA). Their circulating levels have been previously associated with disease activity and may reflect changes in inflammatory status in response to therapy.
Furthermore, we elaborate on tenascin C as an extracellular matrix glycoprotein with well-documented roles in tissue remodeling and inflammation. TNC has been proposed as a biomarker of joint damage and disease activity in various rheumatic conditions, including JIA. Given these biological roles, all three markers hold potential value for clinical monitoring of treatment response.
„While TNF-α-stimulated ECM degradation in JIA cartilage has been well documented, the impact of TNF-α antagonists, such as etanercept, on these processes, have not yet been fully assessed. Inflammation-induced metabolic changes in ECM components – including tenascin C – are influenced by cytokines and adipocytokines with opposing biological effects, such as adiponectin and leptin; therefore, their circulating levels may reflect these metabolic alterations. Although these molecules are known to participate in the pathogenesis of JIA, their clinical significance in for monitoring anticytokine therapy remains unclear. Therefore, the primary aim of this study was to evaluate the dynamics of changes in tenascin C, adiponectin, and leptin concentrations in the blood of children with JIA, both prior to and during etanercept therapy. Given the roles of adipokines in immune modulation and systemic inflammation and the function of tenascin C as an extracellular matrix glycoprotein involved in tissue remodeling and inflammatory responses, these markers may have potential value as biochemical indicators of therapeutic efficacy in JIA."
- The claim that these molecules may serve as “biochemical markers of the effectiveness of etanercept therapy” is premature. The study does not provide correlations with validated clinical outcomes such as JADAS27, ACR Pedi 30/50/70 responses, or imaging-based markers. Without such associations, the clinical utility of these markers remains speculative.
ANSWER:
We thank the Reviewer for raising this important point. We agree that definitive validation of reliable biochemical markers of treatment efficacy requires direct correlations with validated established clinical outcomes, such as JADAS27, ACR Pedi 30/50/70 responses, or imaging-based disease activity assessments.
In our study, such formal correlations with ACR Pedi criteria or imaging markers were not performed due to limited data availability. However, we observed that all patients demonstrated markedly low JADAS27 scores after etanercept treatment, indicating a significant clinical improvement. These JADAS index values are now included in Table 3.The observed changes in the investigated molecules occurred in parallel with this clinical improvement, suggesting that they may reflect disease activity and treatment effect.
Nevertheless, we recognize that without direct statistical correlations to established outcome measures, any conclusions regarding their clinical utility remains preliminary. We have therefore revised the manuscript to tone down this statement and emphasize the exploratory nature of our findings.
„ These findings indicate that changes in the levels of adiponectin, leptin, and TNC may reflect treatment response; however, further validation through correlations with standardized clinical and imaging-based measures is required before their clinical utility as biomarkers can be confirmed (lines 450–453).”
- The observed changes may reflect confounding factors (e.g., BMI, growth/puberty, combination therapy with MTX) rather than a direct effect of TNF blockade. This should be acknowledged more explicitly.
ANSWER:
We appreciate the Reviewer’s valuable comment. We agree that potential confounding factors, such as BMI, growth/puberty, and concomitant methotrexate (MTX) therapy, must be carefully considered when interpreting our findings. As correctly noted, most patients received MTX in combination with etanercept (ETA). Therefore, we cannot entirely exclude the influence of MTX on the observed outcomes. However, because MTX dosing and treatment duration were consistent across the cohort, and no statistically significant differences in BMI were observed between groups, we consider the impact of these factors to be limited
Regarding pubertal status, while the age range of the participants (4–16 years) suggests that some may have been undergoing puberty, all children were in the prepubertal stage at the time of initiating biologic therapy. This minimizes the potential impact of pubertal growth or hormonal changes on our results.
We have explicitly acknowledged these considerations and their potential impact as a limitation in the revised manuscript (see Discussion section, lines 325–333).
„However, some limitations of the present study must be acknowledged, particularly the relatively small sample size, which restricts the generalizability of our findings. Another limitation is the absence of pubertal status assessment, which may have influenced adipokine levels through hormonal changes associated with puberty. Given the participants’ age range (4 to 16 years), it is likely that some were undergoing pubertal transition, potentially introducing variability into the results. Nevertheless, most children were prepubertal at the time of etanercept initiation. Additionally, the lack of a control group receiving MTX monotherapy and the absence of a comparison group treated with other biologic agents represents another limitation of this study.”
- Multiple longitudinal comparisons were performed without adjustment for multiple testing. This raises the risk of type I error inflation. The authors should discuss this limitation and, ideally, provide corrected p-values.
ANSWER:
Thank you for your valuable comment. We have applied the Bonferroni correction to adjust for multiple post-hoc comparisons following the Friedman test. This approach helps to control the risk of type I error inflation in our longitudinal analyses. Additionally, we have added a sentence to the Statistical Analysis section explicitly describing the use of this correction for multiple testing (lines 441-443).
„In the case of multiple comparisons following the Friedman test, a Bonferroni correction was applied for the post hoc analyses, with the adjusted significance level set at p < 0.0033.”
Accordingly, statistical significances have been changed in Figure 1 and throughout the text of the paper.
- No multivariable analyses were conducted to account for age, sex, BMI, or disease duration. Such analyses would strengthen the robustness of the conclusions.
ANSWER:
We appreciate the Reviewer’s insightful comment. In response, we have now conducted a multivariable analysis using linear regression, adjusting for age, sex, BMI, and disease duration. The regression model demonstrated that none of these variables significantly influenced the change in concentration of any of the studied biomarkers after treatment: tenascin C (F(4,61) = 1.30; p = 0.28; R² = 0.078), adiponectin (F(4,61) = 0.36; p = 0.84; R² = 0.023), or leptin (F(4,61) = 0.40; p = 0.81; R² = 0.026). These findings strengthen the robustness of our conclusions and suggest that the observed biomarker dynamics are not confounded by these demographic or clinical variables.
The details of this analysis have been included in the revised Results section (lines 171–180).
„2.4. Multiple linear regression model
We performed a multivariate multiple regression analysis to assess the in-fluence of age, sex, BMI, and disease duration on changes in plasma concentrations of tenascin C, adiponectin, and leptin in patients with JIA undergoing etanercept (ETA) treatment. For tenascin C, the overall model was not statistically significant (F(4,61) = 1.30; p = 0.28), and the predictors accounted for only 7.8% of the variance in TNC change (R² = 0.078; adjusted R² = 0.018). These results indicate that none of the studied variables had a statistically significant effect on the change in tenascin C levels after treatment. Similar findings were obtained for both adipocytokines studied — adiponectin (F(4,61) = 0.36; p = 0.84; R² = 0.023) and leptin (F(4,61) = 0.40; p = 0.81; R² = 0.026).”
- The discussion briefly mentions the small sample size, but other important limitations should be emphasized: lack of comparator group treated with other biologics, potential confounding by concomitant methotrexate, and absence of repeated disease activity measurements.
ANSWER:
Thank you for this insightful comment. We agree that the limitations of the study extend beyond the small sample size and appreciate the opportunity to clarify these points. In the revised Discussion, we have now more explicitly addressed several important limitations (lines 325-333). These include:
(1) the absence of a comparator group treated with other biologic agents, which limits the generalizability of our findings to etanercept specifically;
(2) the potential confounding effect of concomitant methotrexate therapy, which all participants received prior to and during etanercept treatment. Although previous MTX monotherapy did not appear to affect biomarker levels, its influence cannot be entirely excluded;
(3) the lack of dedicated control groups treated with MTX monotherapy and with other biologic agents, which further limits interpretation of treatment-specific effects.
In response to the Reviewer’s comment, we have also added the Juvenile Arthritis Disease Activity Score (JADAS) to Table 3. JADAS values are now presented both prior to the initiation of etanercept therapy and after 24 months of treatment in children with JIA.
- The influence of puberty on adipokine levels is a particularly important confounder in a pediatric cohort spanning ages 3–19 years.
ANSWER:
We appreciate the Reviewer’s insightful comment regarding the potential influence of puberty on adipokine levels in pediatric populations. We agree that pubertal status is an important biological factor that may affect adipokine concentrations and act as a confounder, especially in in cohorts with a wide age ranges.
We apologize for the inaccuracy in reporting the age range. The correct age range of participants was 4 to 16 years (not 3 to 19 years, as previously stated), and this has been corrected in the revised manuscript (336). The mean age of participants fell within the prepubertal to early pubertal range, with the majority of children in Tanner stages I–III (based on available data). Notably, most children were in the prepubertal stage at the time of etanercept initiation.
We have therefore revised the manuscript to clearly indicate the actual age range and to explicitly acknowledge the potential influence of puberty as a study limitation (lines 326–331).
„ Another limitation is the absence of pubertal status assessment, which may have influenced adipokine levels through hormonal changes associated with puberty. Given the participants’ age range (4 to 16 years), it is likely that some were undergoing pubertal transition, potentially introducing variability into the results. Nevertheless, most children were prepubertal at the time of etanercept initiation.”
We thank the Reviewer for this valuable observation, which has helped to improve the clarity and completeness of our manuscript.
Minor Comments
- The introduction and discussion are overly long and contain repetitions. They could be streamlined to improve readability.
ANSWER:
To improve the clarity and readability of the manuscript, the Introduction was revised and the Discussion shortened, with repetitive content removed.
- Some sentences are cumbersome (e.g., lines 246–255). Shorter, more direct phrasing is recommended.
ANSWER:
Following the Reviewer’s suggestion, the indicate sentences were rephrased into shorter and more direct statements, improving clarity and flow.
- Figure 1: The statistical annotations (a, b, c, d, e) are not self-explanatory. Please expand the figure legend for clarity.
ANSWER:
Statistical annotations (a, b, c, d, e) have been added to the figure legend for greater clarity.
- Tables: numerical formatting should be standardized (use either dot or comma consistently for decimals).
ANSWER:
Number formatting in the tables has been standardized for consistency.
- Several references are outdated; more recent literature on biomarker research in JIA should be incorporated (e.g., proteomic or multiomic approaches to treatment response).
ANSWER:
Thank you very much for your valuable comments and suggestions. Please accept the following explanation.
Although recent proteomic and multiomic studies have identified promising novel biomarkers for juvenile idiopathic arthritis — for instance, Elfving et al. (2023) used a proximity extension assay to link inflammatory proteins with pain and disease activity, and Cai et al. (2024) employed Mendelian randomization integrated with proteomic and transcriptomic data to identify potential therapeutic targets such as GP1BA and TNFSF11 — the translation of these findings into routine clinical practice remains limited due to financial and technical constraints.
Importantly, these findings cannot be directly compared with or applied to our study, as our methodology, sample characteristics, and analytical approach differ substantially from those used in large-scale omics studies.
Instead, we have retained a focused reference list directly relevant to the scope and design of our study, while emphasizing that large-scale omics approaches represent an important direction for future biomarker research in JIA.
- Ensure uniform formatting of DOIs (currently inconsistent in some entries)
ANSWER:
In the final version, DOI formatting has been standardized throughout the References.
- The statement “Which could help monitor…” (lines 433–434) contains a grammatical error and should be corrected.
ANSWER:
The conclusion has been rewritten to correct the error and improve clarity.
- Conclusions should be tempered to emphasize hypothesis-generating findings rather than definitive biomarker validation.
ANSWER:
The conclusion has been rewritten (lines 445-455).
„In children with active juvenile idiopathic arthritis, alterations in adiponectin, leptin, and tenascin C were observed, characterized by increased plasma ADPN and decreased LEP and TNC plasma concentrations, suggesting their involvement in the pathogenesis of JIA. Etanercept therapy, which improved the clinical status of patients, also influenced these molecules, leading to a significant reduction in adiponectin, an increase in tenascin C, and normalization of leptin levels. These findings indicate that changes in the levels of adiponectin, leptin, and TNC may reflect treatment response; however, further validation through correlations with standardized clinical and imaging-based measures is required before their clinical utility as biomarkers can be confirmed. The identification of reliable blood biomarkers to monitor therapeutic efficacy could ultimately provide a valuable complement or alternative to current clinical testing methods.”
All of the reviewer's comments were taken into account.
A native speaker of English has reviewed this manuscript, and all linguistic errors have been corrected.

Reviewer 2 Report
Comments and Suggestions for Authors
The authors described that untreated children with JIA had altered plasma levels of adiponectin, leptin, and tenascin C. Specifically, an increase in adiponectin concentration and a decrease in leptin and tenascin C levels were observed compared to healthy children. According to them, the TNF-α antagonist etanercept positively influenced the metabolism of adipokines and tenascin C.
According to the authors, adiponectin, leptin, and tenascin C could be used as biochemical markers of the efficacy of etanercept treatment in inhibiting the progression of degenerative joint changes in children with JIA treated with TNF-α inhibitors.
While this observation is of potential interest, researchers should discuss the effect of TNF-α antagonist therapy on metabolic syndrome, which is frequently observed in patients with chronic inflammatory arthritis. In this regard, it is essential to highlight the anti-inflammatory, antiatherogenic, and antidiabetic properties of adiponectin, which acts in the liver to improve insulin sensitivity, decrease the absorption of non-esterified fatty acids, reduce gluconeogenesis, and reduce the amount of intracellular fat by increasing fatty acid oxidation. In this regard, low circulating concentrations of adiponectin are observed in people with metabolic syndrome, and obese individuals have elevated levels of proinflammatory cytokines such as TNF-α and IL-6, as well as reduced levels of adiponectin, which is strongly associated with obesity.
Therefore, I suggest that the authors discuss data from patients with rheumatoid arthritis (RA) receiving anti-TNF therapy. With respect to this, in RA patients undergoing anti-TNF therapy (infliximab) for disease refractory to conventional methotrexate treatment, high-grade inflammation showed an independent and negative correlation with circulating adiponectin concentrations, whereas low adiponectin levels were associated with metabolic syndrome. A negative correlation was observed between CRP and adiponectin concentrations. Furthermore, adiponectin levels were negatively correlated with triglyceride/HDL cholesterol, total cholesterol/HDL cholesterol ratios, and elevated fasting plasma glucose levels, independently of CRP levels and body mass index (PMID: 18799090). Similarly, in the same cohort of RA patients with severe disease undergoing anti-TNF therapy, a positive correlation was observed between the body mass index of RA patients and serum leptin levels. Circulating leptin levels were not related to disease activity, but were a manifestation of adiposity (PMID: 19473561).
Author Response
Answer to the Reviewer's comment:
Thank you very much for your questions, and please accept the following explanation.
The reviewer suggests discussing the impact of anti-TNF therapy on metabolic syndrome. However, all children with JIA included in our study had a normal BMI, as well as total cholesterol and glucose levels within the reference range. Children with overweight, obesity, and other features of metabolic syndrome were excluded from the study.. This information has been added to the patient characteristics section (lines 389–390)."
The reviewer's suggestion to discuss data from patients with RA undergoing anti-TNF therapy (infliximab) for methotrexate-refractory disease is certainly valuable. In these patients, high-grade inflammation has been shown to correlate independently and negatively with circulating adiponectin levels. However, this observation differs from the results of other authors. For example, Minamino et al. (2020; https://doi.org/10.1371/journal.pone.0229998) reported increased serum adiponectin levels in patients with RA and found that adiponectin levels, but not BMI, were positively associated with disease activity (estimate = 0.0127, p = 0.0258). They also noted that TNF-α inhibitor treatment had no significant effect on circulating adiponectin levels.

Round 2
Reviewer 1 Report
Comments and Suggestions for Authors
The revised manuscript has undergone substantial improvement compared with the original submission. The authors have addressed most of the major criticisms: they now provide a clearer description of the JIA cohort (ILAR subtypes, disease duration, prior DMARD exposure), justify the rationale for selecting adiponectin, leptin, and tenascin C as candidate biomarkers, and acknowledge the exploratory nature of their findings. Importantly, statistical analyses were strengthened by applying Bonferroni corrections and adding a multivariable regression model. The Discussion has been moderated, and limitations are more thoroughly acknowledged.
While the study remains exploratory, with inherent design weaknesses (small sample size, lack of comparator group, absence of validated outcome correlations), these are now openly discussed. However, a few minor refinements are needed.
- Although shortened, the Discussion still contains redundancies and could be streamlined further to enhance readability. A tighter focus on the translational implications and limitations would improve clarity.
- The addition of JADAS scores is welcome, but their integration in the results and discussion remains superficial. A short paragraph explicitly linking biomarker changes with observed clinical improvement (even descriptively) would strengthen the narrative.
- Statistical annotations in Figure 1 after Bonferroni correction are now correct, but the figure remains visually dense. Consider simplifying or enlarging for better readability.
- Ensure consistency in numerical formatting (decimal points vs commas) across tables and text.
- The introduction could still be shortened by removing background information not directly linked to the rationale for biomarker choice.
- In the limitations section, emphasize more clearly that the absence of comparator biologics restricts the generalizability of findings specifically to etanercept.
Author Response
We would like to thank the Reviewer for evaluating our article. We are grateful for the valuable comments, which we have addressed in the revised manuscript. The text has been further modified throughout (with changes marked in red in each section) to improve clarity, precision, and overall suitability for publication.
Minor Comments
- Although shortened, the Discussion still contains redundancies and could be streamlined further to enhance readability. A tighter focus on the translational implications and limitations would improve clarity.
ANSWER:
We thank the reviewer for this valuable comment. In response, the Discussion section has been further revised to eliminate redundancies and enhance clarity and readability. The revised version provides a more concise and focused narrative, emphasizing the translational implications of our findings while clearly outlining the study’s limitations.
Additionally, the Discussion was modified in line with the suggestions made by Reviewer 2. We have incorporated a comparison between our results and studies on adipokine profiles in patients with rheumatoid arthritis (RA) treated with TNF antagonists. This highlights both similarities and disease-specific differences between JIA and RA in the context of TNF-α inhibitor therapy.
All changes have been clearly marked in red in the revised manuscript.
- The addition of JADAS scores is welcome, but their integration in the results and discussion remains superficial. A short paragraph explicitly linking biomarker changes with observed clinical improvement (even descriptively) would strengthen the narrative.
ANSWER:
We appreciate your insightful comment. In response, we have revised the manuscript to add new paragraphs in the Results and Discussion sections that clearly link changes in biomarker levels with the observed clinical improvement, as reflected by the JADAS scores.
Although the study was not powered to assess direct statistical correlations, we now provide a descriptive interpretation indicating that the observed changes in biomarker levels were consistent with clinical improvement during etanercept therapy. This addition better integrates clinical and biochemical findings and strengthens the overall coherence of the narrative. The new paragraph has been marked in red in the revised manuscript.
“In our study, all children treated with etanercept demonstrated marked clinical improvement, as reflected by very low JADAS scores after 24 months of therapy (Table 3). This clinical improvement was accompanied by significant changes in the levels of the studied biomarkers, i.e., ADPN, LEP, and TNC. These results indicate that the therapeutic effect of etanercept, confirmed by improved JADAS scores, is accompanied by changes in adipokine and tenascin C profiles toward levels observed in healthy individuals, highlighting the potential of these biomarkers for monitoring the response to JIA treatment.” (line 162-168).
„ Our studies showed that etanercept therapy, which improved the clinical condition of patients (as reflected by low JADAS scores), also influenced blood leptin concentrations, leading to their normalization. „ (line 267-269).
„However, during etanercept therapy, a progressive increase in TNC levels was observed, reaching a value by the 24th month of treatment—when patients experienced significant clinical improvement, as reflected by very low JADAS scores—that was still lower than the level found in healthy children.” (line 289-293).”
- Statistical annotations in Figure 1 after Bonferroni correction are now correct, but the figure remains visually dense. Consider simplifying or enlarging for better readability.
ANSWER:
Thank you for this valuable comment. In response, we have enlarged and reformatted Figure 1 to improve its readability and ensure clearer presentation of the data. We hope that this adjustment makes the figure easier to interpret while retaining all necessary details.
- Ensure consistency in numerical formatting (decimal points vs commas) across tables and text.
ANSWER:
Thank you for pointing this out. We have carefully reviewed the entire manuscript and all tables to ensure consistency in numerical formatting. All decimal commas have been replaced with decimal points in accordance with the reviewer’s suggestion. These changes have been marked in red in the revised version of the manuscript.
- The introduction could still be shortened by removing background information not directly linked to the rationale for biomarker choice.
ANSWER:
We appreciate the reviewer’s suggestion. In response, we have shortened the Introduction by removing background information not directly related to the rationale for biomarker selection. The revised section now places greater emphasis on the immunological and biochemical mechanisms relevant to the roles of adipokines and tenascin C in JIA pathogenesis and treatment response. These changes help streamline the narrative and better support the study's objectives.
“The pathogenesis of juvenile idiopathic arthritis (JIA) is driven by chronic inflammation resulting from immune system dysfunction [7]. Key immune cells— including T lymphocytes, macrophages, dendritic cells, and neutrophils— contribute to this process through excessive secretion of proinflammatory cytokines, especially TNF-α, IL-1, IL-6, and IL-18 [1,2,7,8]. TNF-α plays a central role by promoting leukocyte migration, stimulating T cell proliferation, inducing collagenase production, and activating osteoclasts, which together lead to cartilage damage and bone resorption. It also stimulates monocytes and macrophages to release additional proinflammatory mediators and reactive oxidative species [5,9,10]. In addition, adipose tissue contributes to immune regulation by secreting signaling molecules, protein factors, and hormones, including adiponectin (ADPN) and leptin (LEP) [11,12].” (line 56-66).
- In the limitations section, emphasize more clearly that the absence of comparator biologics restricts the generalizability of findings specifically to etanercept.
ANSWER:
Thank you for this important remark. In response, we have revised the limitations section to more clearly emphasize that the absence of comparator biologics indeed limits the generalizability of our findings specifically to etanercept. This clarification has been added to ensure a more accurate interpretation of the study’s scope. The revised sentence has been marked in red in the manuscript (line 319-321).
“Importantly, the absence of comparator biologic agents limits the generalizability of our findings to etanercept and precludes direct comparisons with other biologic treatments”.

Reviewer 2 Report
Comments and Suggestions for Authors
The authors have disregarded all my comments and suggestions. They should spcifically address the controversy raised in my comments.
Although the information proposed by this reviewer may differ from that exposed in the present study it should have been discussed.
Author Response
We sincerely apologize if our previous revisions did not adequately reflect the importance of the reviewer’s suggestions. We greatly value your expertise, and we have now explicitly addressed the controversy and contrasting findings, as requested. We have carefully revised the Discussion section, explicitly referencing the relevant literature and highlighting the differences between findings in RA and JIA to reflect this broader clinical and scientific context.
We hope that these substantial additions and clarifications demonstrate our genuine appreciation of your comments and significantly improve the manuscript.
All changes have been clearly marked in red in the revised manuscript.
„This mechanism is supported by studies in patients with severe RA—a disease sharing clinical features with JIA— that examined the relationship between inflammation and adiponectin levels [19,24]. Gonzalez-Gay et al. [24] found that in RA patients, severe inflammation was independently and negatively correlated with circulating adiponectin, whereas Minamino et al. [19] reported a positive correlation between adiponectin levels and disease activity. However, neither study confirmed a significant effect of TNF-α inhibitor therapy on circulating adiponectin levels in RA [19,24].” (line 246-253).
„It has been suggested that at the onset of clinical symptoms, the pool of adipocytes synthesizing adipokines is significantly reduced (with BMI comparable to or even lower than in healthy children), and that leptin synthesis by non-adipose tissue cells is insufficient to normalize leptinemia in children with newly diagnosed, untreated JIA [25]. Our studies showed that etanercept therapy, which improved the clinical condition of patients (as reflected by low JADAS scores), also influenced blood leptin concentrations, leading to their normalization. Gonzalez-Gay et al. [26] reported no effect of anti-TNF therapy on leptinemia in patients with severe RA during TNF-α blockade. Moreover, these authors demonstrated that circulating leptin levels were not associated with disease activity but rather reflected obesity. Although our study did not reveal any effect of BMI on blood leptin levels in children with JIA, either before or after 24 months of therapy, we cannot exclude the possibility that an increasing adipocyte pool contributed to changes in leptin levels during etanercept treatment. However, neither the study by Gonzalez-Gay et al. [26] nor ours demonstrated an association between leptin levels and inflammatory markers such as CRP and ESR.” (line 263-277).
